# Alteration of Relative Rates of Biodegradation and Regeneration of Cervical Spine Cartilage through the Restoration of Arterial Blood Flow Access to Rhomboid Fossa: A Hypothesis

**DOI:** 10.3390/polym13234248

**Published:** 2021-12-03

**Authors:** Kirill V. Zhukov, Alexandre A. Vetcher, Bagrat A. Gasparuan, Alexander Y. Shishonin

**Affiliations:** 1Complementary and Integrative Health Clinic of Dr. Shishonin, 5 Yasnogorskaya Str., 117588 Moscow, Russia; kirizhuk@yandex.ru (K.V.Z.); b.gasparyan@shishonin.ru (B.A.G.); ashishonin@yahoo.com (A.Y.S.); 2Peoples’ Friendship University of Russia (RUDN), 6 Miklukho-Maklaya Str., 117198 Moscow, Russia

**Keywords:** biodegradation rates, arterial hypertension, vertebral cartilage, rhomboid fossa

## Abstract

We found the logical way to prove the existence of the mechanism that maintains the rates of biodegradation and regeneration of cervical spine cartilage. We demonstrate, that after we restore access to arterial blood flow through cervical vertebral arteries to rhomboid fossa it causes the prevalence of regeneration over biodegradation. This is in the frames of consideration of the human body as a dissipative structure. Then the recovery of the body should be considered as a reduction of the relative rates of decay below the regeneration ones. Then the recovery of cervical spine cartilage through redirecting of inner dissipative flow depends on the information about oxygen availability that is provided from oxygen detectors in the rhomboid fossa to the cerebellum. Our proposed approach explains already collected data, which satisfies all the scientific requirements. This allows us to draw conclusions that permit reconsidering the way of dealing with multiple chronic diseases.

## 1. Introduction

Cervical intervertebral disc damage has long been considered a major source of neck pain for quite a long time [1]. Generally, the major reason for such damage is attributed to osteochondrosis [2]. The analysis of accumulated data connects disc degeneration with multiple chronic diseases, including for instance diabetes mellitus type 2 [3]. Namely, for the reason of observed correlations, the recovery of damaged vertebral cartilage is one of the main challenges of modern medicine [4,5]. Therefore, the explanation of reasons for such correlations’ existence plays great importance. The structure of cartilage (mostly extracellular matrix (65–80% liquid content) the dry part of which consists of collagens, proteoglycans, and in smaller amounts lipids, phospholipids, non-collagenous proteins, and glycoproteins [6]) allows the consideration of the process of its decay as biopolymers biodegradation [7]. For that reason, the process of non-invasive integrated recovery could be considered as events with cartilage positioned on the flow (Figure 1).

Since biopolymers are the main part of the dry extracellular matrix, therefore, the cartilage condition could be described in terms of (Vi—Vo). In the outcome of positive value, we observe recovery, while negative—decay. The observation of the recovery of the cartilage on the molecular level confirms that Vi becomes greater than Vo.

In general, there are two approaches to recover damaged vertebral cartilage—surgery [4,8] and integrated treatment [9,10,11]. Nowadays there is a wide variety of techniques for recovery from gene-based and cell-based therapies [12] to a multitude of polymer-based structures [12,13] to replace or support the damaged part of the cartilage. This is a so-called symptomatic treatment that does not change Vi—Vo.

Therefore, the issue of regeneration as an alternative to continuous degeneration eventually attracts the attention of the scientific community [14,15]. The experiments are usually conducted on model animals (e.g., rats [15,16]) that make it possible to measure some macro and molecular parameters of cartilages, analyzing separated tissues at different stages on the level of histological preparations. Such an experimental set usually compares to the averaged data since we could use the same animal only once, not the data from the same object on the different stages of recovery.

The absence of the paired data (before→after) does not provide the ability to conclude about events at the same object, which is of maximal interest. Indeed, only in this case, we will compare apples to apples. Thus, we need to collect data from the same area of the same cartilage just on different stages of the cartilage recovery process without the separation of the cartilage for the histological staining. Moreover, the data, which has been obtained from humans becomes significantly more valuable for Medicine.

What methods give us the best opportunity to observe the level of cartilage deterioration? At present, the most acceptable methods for such data collection are X-ray, CT, and MRI [17]. The analysis of the accumulated data confirms that comparison of the images taken from the same area by X-ray and MRI can exhibit cartilage growth [18].

Below we will connect contemporary observations of cervical cartilage balancing on generation→biodegradation flow [19] with the ideas of homeostasis [20] from the point of view of biological applications of the thermodynamics of irreversible processes [21,22,23].

## 2. Materials and Methods

As a method, we employed a retrospective analysis. As a sample, we used the already collected data from medical records of 2622 patients treated in our Clinic for arterial hypertension (AHT) so far in 2021. The sample consists of 1163 male (66.3 ± 10.2 y/o) and 1459 female (64.1 ± 9.3 y/o) patients. An ultrasound examination of the brachiocephalic arteries was performed twice during the course of therapy in our clinic at the preliminary and final stages. The restoration of the access to arterial blood flow through cervical vertebral arteries to rhomboid fossa was performed according to [24,25]. Therapy started from the manual correction of the cervical intervertebral discs to restore blood flow through the brachiocephalic arteries. The correction is followed by a cycle of 12 visits to corrective exercises leading to the strengthening of the muscular corset of the neck. These visits took from 15 to 41 days. By the end of the course of therapy, BP had been normalized. The measurements of the blood pressure (BP) and linear velocity of arterial blood flow (V_A_)were conducted as described earlier [24] according to the regulations in Russia. The contemporary project, therefore, did not require any consent agreements from human beings. MRI images were taken on 1.5 T Siemens Magnetom Essenza (Siemens AG, Erlangen, Germany) at the MRI Expert Clinic (Moscow, Russia).

## 3. Formulation of the Problem

Contemporary thermodynamics of irreversible processes suggests that every living creature reminds a surfer of the energy flow, and is called a dissipative structure [22,23]. These structures require somehow organized feedback, which can adjust variations in different parameters to keep the structure in certain frames. In chemistry, it causes self-oscillation like Belousov [26] or Sel’kov [27] reactions. Namely, this way of organization of dissipative structure looks for the bystander as homeostasis of a living creature. We need to underline, that homeostasis is maintainable only if parameters do not cross the critical borders that keep the system (dissipative structure) inside the realm where it can still surf [28]. In biological systems the adjustment is centralized. The supreme center of the regulation for vertebrae is located in the cerebellum [29]. Now we would like to understand if it is possible to treat the patient’s herniated cervical cartilage in a way that everything starts to work properly, how can we prove that homeostasis will no longer let cartilage balancing on generation→biodegradation flow, leave the borders, which restrain it from the way of irreversible decay.

## 4. The Explanation of the Hypothesis

We hypothesize that AHT appears as a reaction of oxygen detectors in the brain on the decrement of the oxygen availability in the blood flow to the detector because of the jam, caused by the cervical cartilage damage and the cascade of the processes described extensively below (Figure 2). This cascade resulted in the elevation of blood pressure (BP).

The elevation is a result to attempt to maintain the constant level of (E_CONST_) energy metabolism in the brainstem through energy balance compensation between aerobic (AE) and anaerobic (AN) ways of glycolysis.

The microcirculatory and cellular levels of the AE (oxygen—E_AE_) and AN (glucose, lipoproteins, etc., E_AN_) molecular components of the metabolism are constantly monitored by special brain centers to fulfill
E_CONST_ = E_AE_ + E_AN_(1)

The decrease of E_AE_, due to particular reasons (reduction in oxygen content in the microcirculatory bed and brain stem cells), two types of centralized adaptation reactions take place to maintain the overall unchanged E_CONST_ value. These are reactions of centralized aerobic-anaerobic energy balance compensation (CAAEBC) to maintain the unchanged level of E_CONST_. Moreover, the reactions of AN compensation, as less energy-efficient [30], are triggered only with the complete depletion of the reserves of AE compensation reactions. For the very schematic representation of this energy flow, AE compensation reactions are neurogenic cardiovascular reactions, which are expressed in a steady rise in BP (an increase in the cardiac output force), a narrowing of the peripheral capillaries at rest, and an increase in cardiac rate. The goal of the AE compensation reaction is to increase the brain stem blood perfusion and hence restoration of the E_AE_ level.

AN compensation reactions are neurohumoral metabolic reactions that lead to an increase in the AN metabolism of sugars, phospholipids, and other energetically rich biochemical compounds. The purpose of this reaction is to increase E_AN_, to maintain the balance of E_CONST_, with a reduced E_AE_.

Such reactions of the organism are manifestations of phenotypic adaptation (PA) according to [31]. The PA of the living organism to any changes in the environment first takes place by small forces along a simpler path. First of all, oxygen saturation of the brain occurs to a certain level, after which the reflex mechanism causing compensatory AHT is disconnected. If the brain encounters oxygen starvation for an extended period, then according to the PA theory, changes occur at the biochemical level, namely, the balance of biochemical process shifts, that is, the contribution to the energy balance of AN processes increases and the contribution of AE processes decreases.

The brain, lacking oxygen, determines its decrease, as a decrease in the level of oxygen in the atmosphere and thus, tries to adapt the work of the organism under AE conditions [32,33]. In other words, the brain tries to adapt to the already changed, according to the information from the oxygen detector, external environment, which has remained the same. Since the brain in such a situation begins to receive signals about the critical conditions of the heart, then, as a control center, to save the cardiac resource, it rebuilds the biochemical processes under conditions of a reduced partial pressure of oxygen [34]. There is a shift in AE⇄AN balance towards AN, thus preserving the overall balance of energy that is necessary to fulfill Bauer’s universal principle of biology [35] and to balance the effect on the body of the second law of thermodynamics.

Considering the work of these compensatory mechanisms—“fast” and “slow”, we could present a remarkable clinical example—squeezing the vessels of the neck to cause a person short-term hypoxia (oxygen deprivation) or tough physical exercise (sudden increment of oxygen consumption) will immediately result in the reflexive increase of BP and heart rate [32,33]. When the squeezing or exercise ends, all vital signs will quickly recover back to normal. This is an example of “quick” adaptation.

If a person already has a long-term occlusion of the vessels due to cervical osteochondrosis or there is a narrowing of the lumen of the vessels due to the atherosclerotic process, we will see the manifestations of the action of the “slow” adaptation with a shift in the AE⇄AN balance, namely, the development of the metabolic syndrome, type 2 diabetes [36], and cervical cartilage decay.

Therefore, if we restore the access of blood flow to the oxygen detector, it will lead stepwise, if it is not too late to:Increment of arterial linear blood flow velocity(V_A_) through cervical arteries, since namely they have access to the oxygen detector.Decrement of BP.Restoration of measurable body parameters like pulse, pH, [Fe], etc., to the normal values.Restoration of cartilage, starting from its biopolymeric part, easily visible on MRI.

Let’s see how such a hypothesis allows predicting the work of the behavior of cervical cartilage discs and what measurements should be presented to justify it.

## 5. Discussion on the Hypothesis Verification

Let’s continue with AHT, caused by intervertebral discs compression with hernias and protrusions of the cervical spine. The anatomical features of the cervical vertebrae are such that veins and arteries pass through the holes in their transverse processes (*arteria vertebralis, venae vertebralis*). Due to the displacement of the vertebrae and the constant deep muscle spasm around them, the vessels are subsequently clamped, their lumen is narrowed (the arteries are thin and convoluted), and the blood flow is reduced up to ten times or may even be stopped and, as a result, the amount of delivered oxygen to the oxygen detectors in the brain is dramatically reduced. The systolic peak (PS), end diastolic velocity (ED), maximum and average speed in the cardiac cycle (TAMAX and TAMEAN, respectively), pulsatility index (PI), resistance index (RI) and the systole/diastole (S/D) ratio [37] characterize the recovery of the flow in different extent (Figure 3).

Since the brain through the detectors’ signals observes the lack of oxygen, it takes emergency measures and commands the heart to increase the strength and/or heart rate so that blood, through all the blocks and obstacles, is still able to reach the brain and provide much-needed oxygen. A stable compensatory increase in pressure and/or heart rate is developed—thus the brain is protected from hypoxia. Therefore, as soon as it is possible to unlock the vertebral arteries and veins, the pressure and heart rate should return to normal [32,33]. In this case, if we can heal the patients from AHT through the restoration of vertebral arteries patency and confirm that cervical spine cartilage discs are developing toward restoration, then our hypothesis should be considered as confirmed. The normalization of AHT through the restoration of the blood flow of the brain stem is easy to register by measurement of BP and arterial V_A_ (This preliminary data is collected and exhibited in Table 1).

In this case, the decompression of the vertebral arteries and veins during the correction of deep neck muscles leads to measurable restoration of V_A_ of *sinistra* and *dextra arteria vertebralis* to the normal [38]. The real truth is the fact that with age, there is a progression of osteochondrosis and complications arise that are associated with the gradual displacement of the cervical vertebrae. To prove this hypothesis, we will need to compare data before and after treatment. In Figure 4 the MRI axial images from the same C2/3 disk of a 75.6 y/o patient is demonstrated. The observation proves, that 5 months after the treatment the biodegradation becomes slower than regeneration and allows regeneration to start the restoration of cartilage. According to [39], such changes in MRI allow assuming that changes are cartilage recovery results.

As a rule, for the majority of elderly people, the diagnosis of “essential” or “idiopathic” AHT simply implies nothing more than a compensatory increase in BP due to circulatory disturbances in the brain stem due to compression of the vessels at the level of the cervical spine [40,41,42,43]. Over time the patient develops an osteochondrosis process in the cervical region, and since this is where the vertebral arteries pass through the transverse processes of the vertebrae, their lumen naturally narrows blood flow into the brain stem and into the rhomboid fossa where the vascular center is located. Osteochondrosis is a disease that is directly related to the psychological state of the individual, namely, the accumulation of stress factors in the body. This happened evolutionarily so that any nervous shock manifests itself in the tension of the neck muscles. Therefore, it is necessary to “squeeze the head in the shoulders, hide it” to protect the cervical arteries, because the predator, attacks the neck first. At its core, this reaction is atavism. Evolution should continue and similar atavisms will eventually disappear. However, while there is such an effect, any stress, social or any other kind, produces an automatic reaction—the neck strains, and the head is drawn into the shoulders. Most people have special target muscles that harden when stressed [44]. This is a kind of vicious circle: stress causes the release of adrenaline, which in turn strains the muscles of the neck and upper back, making even more adrenaline, etc.

Soon enough this condition becomes habitual. There is a chronic spasm of deep muscles and in general a spasm of all muscles of the cervical and thoracic sections. Against this background the microcirculation and nutrition of the intervertebral discs and ligaments are disturbed, their weakening occurs, and the vertebrae begin to shift and clamp the vessels. The partial pressure of oxygen in the brain stem decreases because the flow of blood decreases. Due to the activation of the vascular center in the rhomboid fossa, the brain stem delivers an efferent signal to the heart and it starts increasing the pressure by increasing the heart force and the heart rate. According to Dobroborskiy’s theory, this is the first stage of PA. According to the CAAEBC, it is an AN compensation of the energy balance.

That is to say that the central apparatus—the brain—regularly receives less oxygen than it should have received by the parameters that are genetically set as normal [45]. Accordingly, the control center proceeds to the processes of additional energy release from processes not related to breathing (AN).

The brain marks regular overloads and strains on the heart muscle and then activates the mechanism of spasm of the peripheral vessels, thus reducing the burden on the heart. Gradually, a slow centralization of blood circulation takes place. Therefore, the brain temporarily compensates for the state of hypoxia. With the unjustified administration of beta-blockers and vasodilators acting on the renin-angiotensin-aldosterone system, the brain receives an additional toxic effect that nullifies all its adaptive responses. The control center persistently continues the process of compensating the energy balance, due to which, over time, these drugs cease to function properly [46,47,48].

When the process of oxygen starvation is aggravated, the following adaptive mechanism is activated: the biochemical component. There is a need for extracting energy from AN source. There is a shift in AE⇄AN energy balance, and then a shift in the acid-base balance, resulting in body acidosis [49,50,51]. As a result, the picture of PA related to functioning in a medium without oxygen is traced. It looks like diabetes mellitus type 2 could also be developed this way in the form of impaired insulin metabolism and disruption of the normal functioning of the pancreas. We would like to emphasize, that this is only a suggestion and it requires additional data to draw a sturdy conclusion. That is, the organism tries to keep the energy level in the body with the help of these reactions.

If we can replenish the access oxygen to the detector, then the brain stem should reorganize the regulation of biochemical processes from the condition “lack of oxygen” to “normal”. We should be able to observe this process of continuous recovery. In Figure 4 we demonstrated just one of many observations.

## 6. Conclusions

We demonstrated that the restoration of access to the arterial blood flow to the rhomboid fossa leads to:The restoration of the arterial BP;The restoration of the organism’s homeostatic ability to self-repair the cartilage.

The confirmation of the central role of the rhomboid fossa in homeostasis (including BP restoration, cartilage regeneration, etc.).

To additionally prove this hypothesis with objective methods in addition to the data on BP, V_A_ and MRI we will need to provide the data on blood biochemical parameters, e.g., pH, [Fe], etc., to demonstrate the beginning of physiological functions restoration.

## Figures and Tables

**Figure 1 polymers-13-04248-f001:**
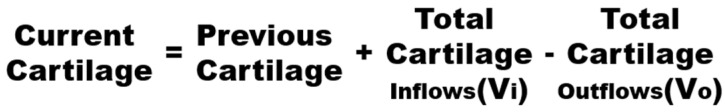
Schematic representation of cartilage on the flow, where Vi is the rate of regeneration and Vo is the rate of biodegradation.

**Figure 2 polymers-13-04248-f002:**
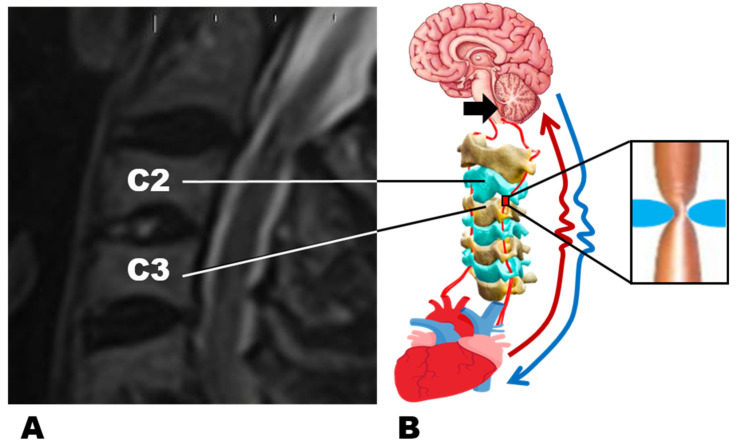
(**A**)—Cervical spine MRI sagittal image demonstrated damaged C2/3 intervertebral disc. The patient is 75.6 y/o. (**B**)—Schematic representation of the caused by the protrusion of intervertebral disc restriction of the access to arterial blood to rhomboid fossa (black arrow).

**Figure 3 polymers-13-04248-f003:**
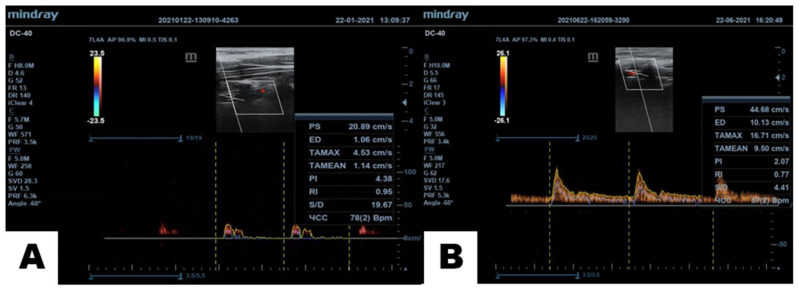
Restoration of V_A_ of a 75.6 y/o patient. (**A**)—V_A_ measurements before and (**B**)—5 months after the treatment according to [24,25]. The 10 fold increment of diastolic V_A_ (ED) and nine fold increment of TAMEAN confirm the opening of the access to rhomboid fossa.

**Figure 4 polymers-13-04248-f004:**
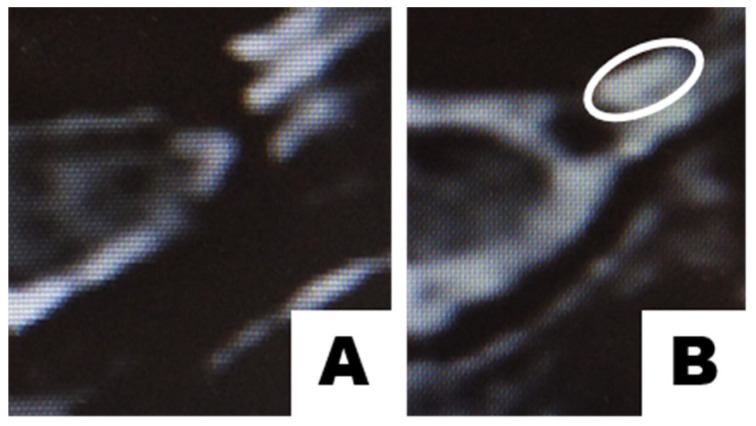
(**A**)—Cervical spine MRI axial image of C2/3 intervertebral disc before and (**B**)—5 months after the treatment according to [24,25]. The area of the prevalance of the regeneration over biodegradation is surrounded by the white line.

**Table 1 polymers-13-04248-t001:** Results on the restoration of vertebral arteries V_A_ by cervical cartilage recovery according to [24,25]. The data was collected for the entire sample.

Parameter	Patients Number	% of Total
Lowering BP on 10–20 torr	740 *	28
Lowering BP on 20–40 torr	1604 *	61
Lowering BP for ≥40 torr	278 *	11
Pulse normalization	2342	89
Increment of V_A_ ≥ 25%	2622	100

* 740 + 1604 + 278 = 2622, so the BP-lowering experienced 100% of the patients.

## Data Availability

As a method, we employed a retrospective analysis and we are able to provide the employed data with the exception of the part, that is covered by the The Russian Federal Law on Personal Data (No. 152-FZ).

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
