# Peer review of "Alteration of Relative Rates of Biodegradation and Regeneration of Cervical Spine Cartilage through the Restoration of Arterial Blood Flow Access to Rhomboid Fossa: A Hypothesis"

_polymers, 2021, doi:10.3390/polym13234248_

Round 1

Reviewer 1 Report

The manuscript has improved a lot after revision, therefore i am accepting this manuscript in present form.

Author Response

Dear Reviewer:

Thank you so much for your efforts to give our submission the best shape for the Readers.

Reviewer 2 Report

Review of the manuscript “Alteration of relative rates of biodegradation and regeneration of cervical spine cartilage through the restoration of arterial blood flow acces to rhomboid foss: a hypothesis”. By K. V, Zhukov, A. Vetcher, B.A. Gasparuan, A.Y. Shishonin.

The objective of the paper is to demonstrate that there is a relationship among the decrease of blood flow through cervical vertebral arteries, as consequence of damaged cartilage, and the increase in arterial hypertension and the development of chronic diseases as type 2 diabetes. The authors want to prove this hypothesis using measurements of blood pressure and linear velocity of arterial blood flow from medical records of patients that have received some cartilage recovery treatment.

At this moment, the article cannot be considered for publication. It is not well written and It needs extensive  language polishing.

 In addition, the materials and methods section should be improved including the necessary  information from 24 and 25 references.

Author Response

Response for

Review of the manuscript “Alteration of relative rates of biodegradation and regeneration of cervical spine cartilage through the restoration of arterial blood flow acces to rhomboid foss: a hypothesis”.

Dear Reviewer:

Thank you so much for your efforts to make Polymers better. As for your comments:

The objective of the paper is to demonstrate that there is a relationship among the decrease of blood flow through cervical vertebral arteries, as consequence of damaged cartilage, and the increase in arterial hypertension and the development of chronic diseases as type 2 diabetes. The authors want to prove this hypothesis using measurements of blood pressure and linear velocity of arterial blood flow from medical records of patients that have received some cartilage recovery treatment.

Technically relationship between the decrease of blood flow through cervical vertebral arteries and the increase in arterial hypertension is the fact [1, 2] (ref. [32,33]). We just hypothesized that the possible reason for it is the restriction of the blood flow access to the rhomboid fossa. From our point of view, the recovery eventually leads to the regeneration of the biopolymeric part of the cartilage, which is observed by MRI in some cases. We regret that this idea has somehow vanished from the focus. And we rewrite the discussion part to make it clear. This submission is not devoted to type 2 diabetes, which is mentioned just as an example of the diseases, that could be autocorrected, according to our point of view, if the hypothesis works. The type 2 diabetes part definitely required additional experiments and could be excluded, but we prefer to leave it.

We marked all of the changed parts yellow, since the version, that we uploaded from the Polymers’ website doesn’t allow us to track the corrections.

At this moment, the article cannot be considered for publication. It is not well written and It needs extensive language polishing.

We carefully rewrote the Discussion section and invite a native American speaker to suggest how to increase readers’ understanding.  

In addition, the materials and methods section should be improved including the necessary information from 24 and 25 references.

We expanded the information from ref. [24,25]

  1. Curtelin D, Morales-Alamo D, Torres-Peralta R, Rasmussen P, Martin-Rincon M, Perez-Valera M, Siebenmann C, Perez-Suarez I, Cherouveim E, Sheel AW et al: Cerebral blood flow, frontal lobe oxygenation and intra-arterial blood pressure during sprint exercise in normoxia and severe acute hypoxia in humans. J Cereb Blood Flow Metab 2018, 38(1):136-150.
  2. He ZB, Lv YK, Li H, Yao Q, Wang KM, Song XG, Wu ZJ, Qin X: Atlantoaxial Misalignment Causes High Blood Pressure in Rats: A Novel Hypertension Model. Biomed Res Int 2017, 2017:5986957.

Round 2

Reviewer 2 Report

I think that the authors have taken into account my suggestions and that the manuscript can be published. There is only a minor detail to change: font size of text in Figure 1 should be reduced.
